# On interaction between augmentations and corruptions in natural corruption robustness

## Abstract

Invariance to a broad array of image corruptions, such as warping, noise, or color shifts, is an important aspect of building robust models in computer vision. Recently, several new data augmentations have been proposed that significantly improve performance on ImageNet-C, a benchmark of such corruptions. However, there is still a lack of basic understanding on the relationship between data augmentations and test-time corruptions. To this end, we develop a feature space for image transforms, and then use a new measure in this space between augmentations and corruptions called the Minimal Sample Distance to demonstrate there is a strong correlation between similarity and performance. We then investigate recent data augmentations and observe a significant degradation in corruption robustness when the test-time corruptions are sampled to be perceptually dissimilar from ImageNet-C in this feature space. Our results suggest that test error can be improved by training on perceptually similar augmentations, and data augmentations may risk overfitting to the existing benchmark. We hope our results and tools will allow for more robust progress towards improving robustness to image corruptions.

## 1 Introduction

Robustness to distribution shift, *i.e.* when the train and test distributions differ, is an important feature of practical machine learning models. Among many forms of distribution shift, one particularly relevant category for computer vision are image corruptions. For example, test data may come from sources that differ from the training set in terms of lighting, camera quality, or other features. Post-processing transforms, such as photo touch-up, image filters, or compression effects are commonplace in real-world data. Models developed using clean, undistorted inputs typically perform dramatically worse when confronted with these sorts of image corruptions (Hendrycks & Dietterich, 2018; Geirhos et al., 2018). The subject of corruption robustness has a long history in computer vision (Simard et al., 1998; Bruna & Mallat, 2013; Dodge & Karam, 2017) and recently has been studied actively with the release of benchmark datasets such as ImageNet-C (Hendrycks & Dietterich, 2018).

One particular property of image corruptions is that they are low-level distortions in nature. Corruptions are transformations of an image that affect structural information such as colors, textures, or geometry (Ding et al., 2020) and are typically free of high-level semantics. Therefore, it is natural to expect that *data augmentation* techniques, which expand the training set with random low-level transformations, can help with learning robust models. Indeed, data augmentation has become a central technique in several recent methods (Hendrycks et al., 2019; Lopes et al., 2019; Rusak et al., 2020) that achieve large improvements on ImageNet-C and related benchmarks.

One caveat for data augmentation based approaches is the test corruptions are expected to be *unknown* at training time. If the corruptions are known, they may simply be applied to the training set as data augmentations to trivially adapt to the test distribution. Instead, an ideal robust model needs to be robust to *any* valid corruption, including ones unseen in any previous benchmark. Of course, in practice the robustness of a model can only be evaluated approximately by measuring its corruption error on a representative corruption benchmark. To avoid trivial adaptation to the benchmark, recent works manually exclude test corruptions from the training augmentations. However, with a toy experiment presented in Figure 1, we argue that this strategy alone might not be enough and that visually similar augmentation outputs and test corruptions can lead to significant benchmark improvements even if the exact corruption transformations are excluded.

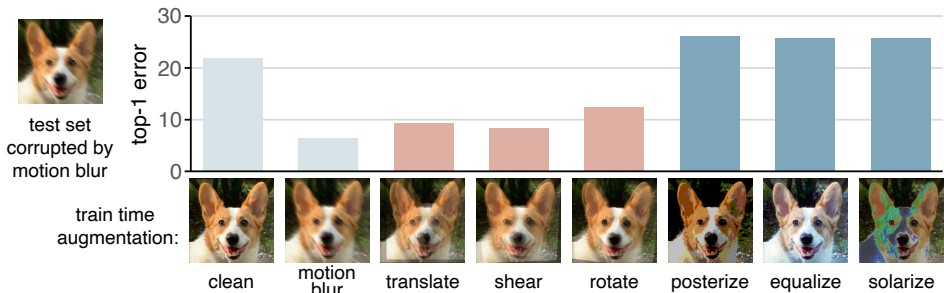

Figure 1: **A toy experiment.** We train multiple models on CIFAR-10 (Krizhevsky et al., 2009) using different augmentation schemes. Each scheme is based on a single basic image transformation type and enhanced by overlaying random instantiations of the transformation for each input image following Hendrycks et al. (2019). We compare these models on the CIFAR-10 test set corrupted by the motion blur, a corruption used in the ImageNet-C corruption benchmark Hendrycks & Dietterich (2018). None of the augmentation schemes contains motion blur; however, the models trained with geometric-based augmentations significantly outperform the baseline model trained on the clean images while color-based augmentations show no gains. We note the geometric augmentations can produce a result visually similar to a blur by overlaying copies of shifted images[1].

This observation raises two important questions. One, *how exactly does the similarity between train time augmentations and corruptions of the test set affect the error?* And two, if the gains are due to the similarity, the improvements may not translate into better robustness to other possible corruptions, so *do we ever risk overfitting existing corruption benchmarks using a new augmentation scheme?* In this work, we take a step towards answering these questions, with the goal of better understanding the relationship between data augmentation and test-time corruptions. Using a feature space on image transforms and a new measure called Minimal Sample Distance (MSD) on this space, we are able to quantify the distance between augmentation schemes and classes of corruption transformation. With our approach, we empirically show an intuitive yet surprisingly overlooked finding:

*Augmentation-corruption perceptual similarity is a strong predictor of corruption error.*

Based on this finding, we perform additional experiments to show that data augmentation aids corruption robustness by increasing perceptual similarity between a (possibly small) fraction of the training data and the test set. To further support our claims, we introduce a set of new corruption, called CIFAR/ImageNet-$\bar{\text{C}}$, to test the degree to which common data augmentation methods overfit original the CIFAR/ImageNet-C. To choose these corruptions, we expand the set of natural corruptions and sample new corruptions that are far away from CIFAR/ImageNet-C in our feature space for measuring perceptual similarity. We then demonstrate that augmentation schemes designed specifically to improve robustness show significantly degraded performance on CIFAR/ImageNet-$\bar{\text{C}}$. Some augmentation schemes still show some improvement over baseline, which suggests meaningful progress towards general corruption robustness is being made, but different augmentation schemes exhibit different degrees of generalization capability. As an implication, caution is needed for fair robustness evaluations when additional data augmentation is introduced.

These results suggest a major challenge that is often overlooked in the study of corruption robustness: *overfitting indeed occurs*. Since perceptual similarity can predict performance, for any fixed finite set of test corruptions, improvements on that set may generalize poorly to dissimilar corruptions. However, perceptual similarity is not expected to be the only interaction between augmentations and corruptions, so a proposed augmentation scheme's degree of generalization capability may not be immediately clear. We hope that our results, together with new tools and benchmarks, will help researchers better understand *why* a given augmentation scheme has good corruption error and whether it should be expected to generalize to dissimilar corruptions. On the positive side, our experiments show that *generalization does emerge* within perceptually similar classes of transform, and that only a *small fraction* of sampled augmentations need to be similar to a given corruption. Section 6 discusses these points in more depth.

---

[1]Example transforms are for illustrative purpose only and are exaggerated. Base image © Sehee Park.

## 2 RELATED WORK

**Corruption robustness benchmarks and analysis.** ImageNet-C (Hendrycks & Dietterich, 2018) is a corruption dataset often used as a benchmark in robustness studies. Other corruption datasets (Hendrycks et al., 2020; Shankar et al., 2019) collect corrupted images from real world sources and thus have a mixture of semantic distribution shifts and perceptual transforms. Corruption robustness differs from adversarial robustness (Szegedy et al., 2014), which seeks invariance to small, worst case distortions. One notable difference is that improving corruption robustness often slightly improves regular test error, instead of harming it. Yin et al. (2019) analyzes corruption robustness in the context of transforms' frequency spectra; this can also influence corruption error independently from perceptual similarity. Dao et al. (2019); Wu et al. (2020) study the theory of data augmentation for regular test error. Hendrycks et al. (2020); Taori et al. (2020) study how the performance on synthetic corruption transforms generalizes to performance on corruption datasets collected from the real world. Here we do not address this issue directly but touch upon it in the discussion.

**Improving corruption robustness.** Data augmentations designed to improve robustness include AugMix (Hendrycks et al., 2019), which composites common image transforms, Patch Gaussian (Lopes et al., 2019), which applies Gaussian noise in square patches, and ANT (Rusak et al., 2020), which augments with an adversarially learned noise distribution. AutoAugment (Cubuk et al., 2019) learns augmentation policies that optimize clean error but has since been shown to improve corruption error (Yin et al., 2019). Mixup (Zhang et al., 2018a) can improve robustness (Lee et al., 2020), but its label augmentation complicates the dependence on image augmentation. Stylized-ImageNet (Geirhos et al., 2019) can improve robustness, but it is at the cost of clean test error without additional fine-tuning. Noisy Student (Xie et al., 2020) and Assemble-ResNet (Lee et al., 2020) combine data augmentation with new models and training procedures and greatly enhance corruption robustness.

## 3 PERCEPTUAL SIMILARITY FOR AUGMENTATIONS AND CORRUPTIONS

First, we study the importance of similarity between augmentations and corruptions for improving performance on those corruptions. To do so, we need a means to compare augmentations and corruptions. Both types of transforms are perceptual in nature, meaning they affect low-level image structure while leaving high-level semantic information intact, so we expect a good distance to be a measure of *perceptual similarity*. Then, we need to find the appropriate measure of distance between the augmentation and corruption *distributions*. We will argue below that distributional equivalence is not appropriate in the context of corruption robustness, and instead introduce the *minimal sample distance*, a simple measure that does capture a relevant sense of distribution distance.

**Measuring similarity between perceptual transforms.** We define a perceptual transform as a transform that acts on low-level image structure but not high-level semantic information. As such, we expect two transforms should be similar if their actions on this low-level structure are similar, independent of algorithmic or per-pixel differences between them. A closely related, well-studied problem is the perceptual similarity between *images*. A common approach is to train a neural network on a classification task and use intermediate layers as a feature space for measuring distances (Zhang et al., 2018b). Here we adapt this idea to instead obtain a feature space for measuring distances between perceptual transforms.

We start with a feature extractor for images, which we call $\hat{f}(x)$. To train the model from which we will extract features, we assume access to a dataset $\mathbb{D}$ of image label pairs $(x, y)$ associated with a classification task. The model should be trained using only default data augmentation for the task in question so that the feature extractor is independent of the transforms we will use it to study. In order to obtain a very simple measure, we will use just the last hidden layer of the network as a feature space.

A perceptual transform $t(x)$ may be encoded by applying it to all images in $\mathbb{D}$, encoding the transformed images, and averaging the features over these images. For efficiency, we find it sufficient to average over only a randomly sampled subset of images $\mathbb{D}_S$ in $\mathbb{D}$. We show in Appendix C this produces stable estimates for reasonable numbers of images. The random choice of images is a property of the feature extractor, and so remains fixed when encoding multiple transforms. This reduces variance when computing distances between two transforms. The transform feature extractor

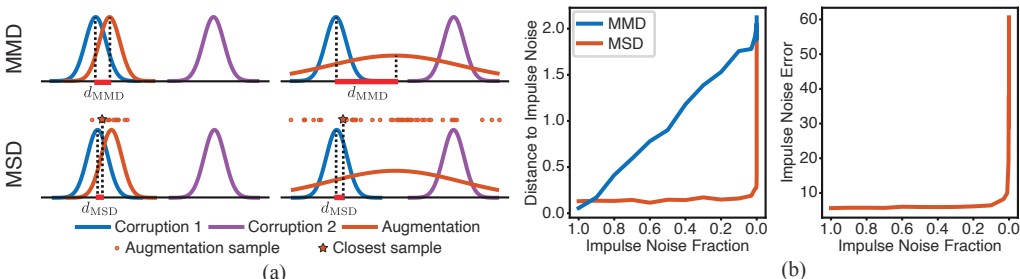

Figure 2: (a) Schematic comparison of MMD to MSD. MMD measure between distribution centers and is only small if the augmentation overlaps with a corruption. MSD measures to the nearest sampled point in set of samples (marked by a star), and is small even for broad distributions that overlap with multiple corruptions. (b) We test on images corruption with *impulse noise*, and train on images augmented with a mixture of *impulse noise* and *motion blur*. As the mixing fraction of *impulse noise* decreases, MMD between the augmentation and corruption grows linearly while MSD and error stay low until nearly 0% mixing fraction.

is given by $f(t) = \mathbb{E}_{x \in \mathbb{D}_S}[\hat{f}(t(x)))]$. The *perceptual similarity* between an augmentation and a corruption can be taken as the $L_2$ distance on this feature space $f$.

**Minimal sample distance.** We now seek to compare the distribution of an augmentation scheme $p_a$ to a distribution of a corruption benchmark $p_c$. A simple first guess would be to measure how close to equivalent the distributions are. Indeed, if the goal was to optimize error on a *known* corruption distribution, exact equivalence of distributions is the correct measure to minimize. But since the goal is robustness to general, *unknown* corruption distributions, a good augmentation scheme should be equivalent to no single corruption distribution.

To illustrate this behavior, consider a toy problem where we have access to the corruption transforms at training time. A very rough, necessary but insufficient measure of distributional similarity is $d_{\mathrm{MMD}}(p_a, p_c) = ||\mathbb{E}_{a \sim p_a}[f(a)] - \mathbb{E}_{c \sim p_c}[f(c)]||$. This is the maximal mean discrepancy on a fixed, finite feature space, so for brevity we will refer to it as MMD. We still employ the featurization $f(t)$, since we are comparing transforms and not images, unlike in typical domain adaptation. Consider two corruption distributions, here *impulse noise* and *motion blur*, and an augmentation scheme that is a mixture of the two corruption distributions. Figure 2b shows that MMD between the augmentation and *impulse noise* corruption scales linearly with mixing fraction, but error on *impulse noise* remains low until the mixing fraction is almost 0% impulse noise. This implies distributional similarity is a poor predictor corruption error. Indeed, in the context of corruption robustness, low $d_{\mathrm{MMD}}$ with any one corruption distribution is likely bad thing: it suggests the augmentation scheme overfits that one corruption distribution at expense of performance on other, dissimilar corruption distributions.

Our expectation for the behavior of the error in Figure 2b is that networks can often successfully memorize rare examples seen during training, so that only a very small fraction of sampled images need *impulse noise* augmentations to perform well on *impulse noise* corruptions. An appropriate distance should then measure how close augmentation samples can come to the corruption distribution, even if the density of those samples is low. We thus propose a very simple measure called *minimal sample distance (MSD)*, which is just the perceptual similarity between an average corruption and the closest augmentation from a finite set of samples $\mathbb{A} \sim p_a$:

$$d_{\mathrm{MSD}}(p_a, p_c) = \min_{a \in \mathbb{A} \sim p_a} ||f(a) - \mathbb{E}_{c \sim p_c}[f(c)]||. \tag{1}$$

A schematic comparison of MMD and MSD is shown in Figure 2a. While both MMD and MSD are small for an augmentation scheme that is distributionally similar to a corruption distribution, only MSD remains small for a broad distribution that occasionally produces samples near multiple corruption distributions. Figure 2b shows MSD, like test error, remains small for most mixing fractions in the toy problem described above. Note that the need for our measure to accommodate robustness to general, unknown corruption distributions has led it to be asymmetric, so it differs from more formal distance metrics that may be used to predict generalization error, such as the Wasserstein distance (Zilly et al., 2019).

## 4    PERCEPTUAL SIMILARITY IS PREDICTIVE OF CORRUPTION ERROR

We are now equipped to measure how important this augmentation-corruption similarity is for corruption error. For a large number of augmentation schemes, we will measure both the MSD to a corruption distribution and the corruption error of a model trained with that scheme. We will find a correlation between MSD and corruption error, which provides evidence that networks can successfully generalize across perceptually similar transforms. Then, we will calculate the MSD for augmentation schemes in the literature that have been shown to improve error on corruption benchmarks. We will find a correlation between MSD and error here as well, which suggests the success of these models is in part explained by their perceptual similarity to the benchmark. This implies there may be a risk that the augmentation schemes overfit the benchmark, since we would not expect this improvement to transfer to a dissimilar corruption.

### 4.1    EXPERIMENTAL SETUP

**Corruptions.** We use CIFAR-10-C (Hendrycks & Dietterich, 2018), which is a common benchmark used for studying corruption robustness. It consists of 15 corruptions, each further split into five different severities of transformation, applied to the CIFAR-10 test set. The 15 corruptions fall into four categories: per-pixel noise, blurring, synthetic weather effects, and digital transforms. We treat each corruption at each severity as a separate distribution for the sake of calculating MSD and error; however, for simplicity we average errors and distances over severity to present a single result per corruption. Examples of each corruption are shown in Figure 13 in Appendix E.

**Space of augmentation schemes.** To build each sampled augmentation transform, we will composite a set of base augmentations. For base augmentations, we consider the nine common image transforms used in Hendrycks et al. (2019), shown in Figure 12 of Appendix E. There are five geometric transforms and four color transforms. By taking all subsets of these base augmentations, we obtain $2^9 = 512$ unique augmentation schemes, collectively called the *augmentation powerset*. Also following Hendrycks et al. (2019), we composite transforms in two ways: by applying one after another, or by applying them to copies of the image and then linearly superimposing the results.

**Computing similarity and corruption error.** A WideResNet-40-2 (Zagoruyko & Komodakis, 2016) model is pre-trained on CIFAR-10 using default augmentation and training parameters from Hendrycks et al. (2019). WideResNet is a common baseline model used when studying data augmentation on CIFAR-10 (Hendrycks et al., 2019; Cubuk et al., 2019; Zhang et al., 2018a). Its last hidden layer is used as the feature space. For MSD, we average over 100 images, 100 corruptions, and minimize over 100k augmentations. We argue in Appendix C that these are reasonable choices. Images are from the training set and do not have default training augmentation. For corruption error evaluation, we also use a WideResNet-40-2 and the same training parameters.

### 4.2    ANALYSIS

**MSD correlates with corruption error.** First, we establish the correlation between MSD and corruption error on the augmentation powerset. Figure 3 shows the relationship between distance and corruption error for both MSD on four example corruption distributions. MSD shows strong correlation with corruption error across corruptions types in all four categories of CIFAR-10-C. In Figure 14 in Appendix F, we compare to MMD and confirm that MMD correlates poorly with corruption error, as expected. In particular, our expectation is that broad augmentation schemes with many base transforms produce samples similar to a larger set of corruptions, even if those samples occur less frequently. This leads to both lower MSD and lower corruption error but higher MMD. Additionally, the correlation between MSD and corruption error suggests that perceptual similarity is a predictor of corruption error. However, as shown in Table 1, most but not all corruptions show strong correlation between MSD and error: 12 of 15 have Spearman rank correlation greater than 0.6. A complete set of correlation plots is shown in Figure 15 in Appendix F.

**An example of perceptual similarity.** Here we briefly illustrate the perceptual nature of the similarity measure, using an example with two base augmentations and two corruptions. The augmentation *solarize* and the corruption *impulse noise* both insert brightly colored pixels into the image, though in different ways. The augmentation *x translation* and the corruption *motion blur* are both geometric transforms, and linear superpositions of *x translation* are visually similar to blurring. Examples of

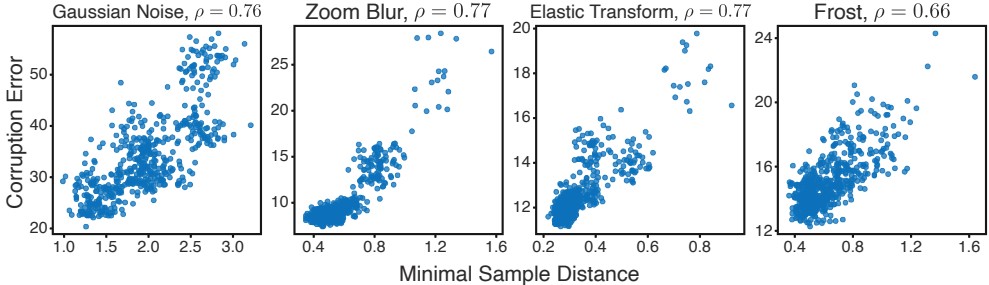

Figure 3: Example relationships between MSD and corruption error for different corruptions. $\rho$ is the Spearman rank correlation. MSD correlates well with error across all four categories of corruption in CIFAR-10-C.

Table 1: Spearman's rank coefficient for the correlation between MSD and corruption error. Correlations are high for most corruptions, including those in all four categories in ImageNet-C. However, *brightness*, *pixelate*, and *JPEG compression* show poor correlation.

| Corruption | $\rho$ | Corruption | $\rho$ | Corruption | $\rho$ |
|---|---|---|---|---|---|
| Gaussian Noise | 0.76 | Zoom Blur | 0.77 | Snow | 0.65 |
| Shot Noise | 0.83 | Glass Blur | 0.69 | Contrast | 0.66 |
| Impulse Noise | 0.90 | Brightness | 0.27 | Pixelate | 0.35 |
| Motion Blur | 0.86 | Fog | 0.68 | JPEG Compression | 0.33 |
| Defocus Blur | 0.83 | Frost | 0.66 | Elastic Transform | 0.77 |

these transforms are shown in Figure 12 and Figure 13 in Appendix E. Figure 4 shows MSD vs error where augmentation schemes that include *solarize* and *x translation* are colored. It is clear that including an augmentation greatly decreases MSD to its perceptually similar corruption, while having little effect on MSD to its perceptually dissimilar corruption.

**MSD and corruption error in real augmentation methods.** The augmentation powerset may be used as a baseline for comparing real data augmentation schemes. Figure 5 shows example MSD-error correlations for Patch Gaussian, AutoAugment, and Augmix, along with the cloud of augmentation powerset points. The real augmentation schemes follow the same general trend that lower error predicts lower MSD. A few intuitive correlations are also captured in Figure 5. Patch Gaussian has low MSD to corruptions with noise corruptions and to *glass blur* which introduces random pixel-level permutations as noise, and may be distributionally similar to the noise corruptions, as we argue in Appendix A. AutoAugment, which contains contrast and Gaussian blurring augmentations in its sub-policies, has low MSD with *contrast* and *defocus blur*. AugMix, which introduces fewer base augmentations that AutoAugment but composites them in ways that may produce perceptually new effects (such as blurring from superimposing translations), has low MSD with all corruptions.

The fact that improved corruption error typically implies greater similarity suggests overfitting may be an issue in the study of corruption robustness. For two augmentation schemes compared on a

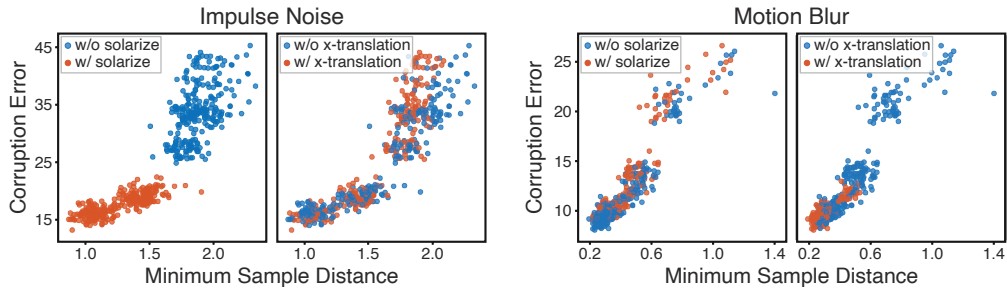

Figure 4: Example relationships between base augmentations and corruptions. Including 'solarize' reduces MSD on the perceptually similar *impulse noise* corruption, while including *x translation* reduces MSD on the perceptually similar *motion blur* corruption. MSD is not decreased for dissimilar augmentation-corruption pairs.

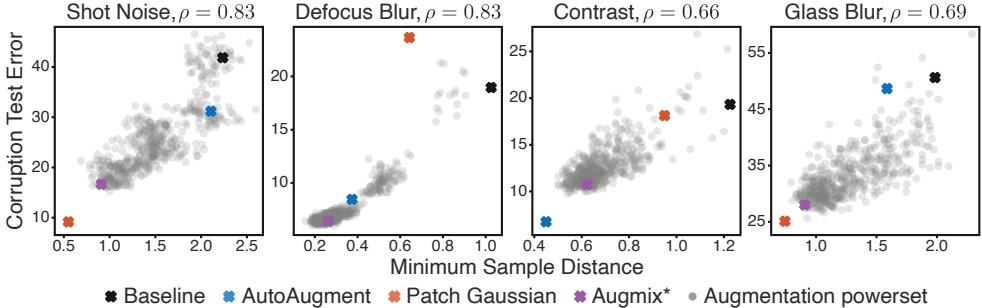

Figure 5: Example correlations for augmentation schemes from the literature. Patch Gaussian is similar to noise, while AutoAugment is similar to contrast and blur, as expected from their formulation. Glass blur acts more like a noise corruption than a blur for these augmentation schemes, likely because it randomly permuting pixels. *AugMix here refers to just the augmentation distribution in Hendrycks et al. (2019), not the proposed Jensen-Shannon divergence loss, which gives additional improvements in corruption error.

fixed corruption benchmark, the one that is more perceptually similar to the corruptions may perform better, but also generalize to dissimilar corruptions worse. This perceptual similarity may be exact and explicit, such as between AutoAugment, which contains *contrast* augmentations, and the *contrast* corruption. But it may also be naively unexpected, such as the similarity between Patch Gaussian and *glass blur* (since *glass blur* shares similarity with the noise corruptions) or between AugMix and blurs (since superimposed translations are blur-like). Note, however, that perceptual similarity is not expected to be the only interaction between augmentations and corruptions (for instance, the frequency dependence of Yin et al. (2019) is another), and other interactions may or may not lead to good generalization. Thus, tools such as MSD help us determine *why* an augmentation scheme improves corruption error, so we can better analyze and understand if newly proposed methods will generalize beyond their tested benchmarks. In the next section, we test this generalization directly by finding corruptions that are dissimilar to ImageNet-C.

## 5 IMAGENET-$\overline{\text{C}}$: BENCHMARKING WITH DISSIMILAR CORRUPTIONS

We now introduce a set of corruptions, called ImageNet-$\overline{\text{C}}$, that are perceptually dissimilar to ImageNet-C in our transform feature space and will show that several augmentation schemes have degraded performance on the new dataset, suggesting that they have indeed overfit ImageNet-C to at least some degree. We emphasize that the dataset selection method does not involve any augmentation scheme beyond the default one used to train the feature extractor and was fixed before we looked at the results for different augmentations, so there should be no inadvertent adversarial selection against the augmentation schemes.

**Dataset construction.** An overview of the dataset construction is presented here, with specific details described in Appendix D.1. We construct a set of 30 new corruptions types in 10 severities, from which the 10 most dissimilar corruptions types will be chosen. These corruptions are adapted from common filters and noise distributions available online (Huxtable, 2006; Gladman, 2016) and are designed to produce human interpretable images. The transforms include warps, blurs, color distortions, noise additions, and obscuring effects. Examples are shown in Appendix E.

To assure that the new dataset is no harder than ImageNet-C, we restrict the average corruption error of the new dataset to be similar to that of ImageNet-C for default augmentation. We then generate many potential datasets and measure the average shift in distance to ImageNet-C that each corruption contributes, shown in Figure 16 in Appendix F. Note that while MSD was used to establish a correlation between perceptual similarity and error for augmentations and corruptions, here we are comparing corruptions to other corruptions and thus use MMD as the measure of distance in our transform feature space. Some generally intuitive results are recovered. First, *scatter* and *blue noise*, which are conceptually similar to ImageNet-C's *glass blur* and *gaussian noise* corruptions, consistently lead to more similarity with ImageNet-C. Second, lower-frequency noise distortions tend to be included in more dissimilar datasets, possibly in contrast to ImageNet-C's high frequency pixel noise. However, *blue noise sample*, which shares some visual similarities with *impulse noise*,

Figure 6: Example CIFAR-10-$\overline{\text{C}}$ and ImageNet-$\overline{\text{C}}$ corruptions. While still human interpretable, new corruptions are sampled to be dissimilar from CIFAR-10/ImageNet-C. Base images © Sehee Park and Chenxu Han.

also leads to larger distances. ImageNet-$\overline{\text{C}}$ then consists of the 10 corruptions types with the largest average shift in distance. Like ImageNet-C, each is included in five different severities, with severities chosen so that the average error matches ImageNet-C for default augmentation. Example transforms from ImageNet-$\overline{\text{C}}$ and CIFAR-10-$\overline{\text{C}}$ are shown in Figure 6.

**Results.** We test AutoAugment, Patch Gaussian, AugMix, and ANT[3x3] on our new datasets and show results in Table 2. CIFAR-10 models are WideResNet-40-2 with training parameters from Hendrycks et al. (2019), ImageNet (Deng et al., 2009) models are ResNet-50 (He et al., 2016) with training parameters from Goyal et al. (2017). Models use default data augmentation as well as the augmentation being tested, except ImageNet color jittering is not used. All corruptions are applied in-memory, instead of loaded from a compressed file; this can affect results especially on high frequency corruptions. Given the evidence of overfitting from Section 4, we expect these methods to have worse error on the new corruptions. Indeed, every augmentation scheme performs worse, even when baseline improves slightly.

Additionally, there are a few other suggestive patterns. First, intuitively broader augmentation schemes perform better: AugMix and AutoAugment degrade less than Patch Gaussian or ANT. Patch Gaussian in particular, which has been identified as perceptually similar to pixel-noise type corruptions alone, sees a particularly large drop in performance. Second, AutoAugment is the only tested augmentation scheme that was designed before the release of ImageNet-C, and it has the smallest performance drop, despite known overlaps with the *brightness*, *contrast*, and *defocus blur* corruptions in ImageNet-C. This suggests that having ImageNet-C on hand may make it difficult to avoid overfitting to. Finally, it is possible that *blue noise sample* is a failure mode of our distance: it both shares visual features with *impulse noise* and most data augmentations have low error on it. On other corruptions, the increase in error is even worse than the mean would suggest, and even broad augmentations like AugMix are no better than baseline on several individual corruptions.

Despite this degradation, note that the data augmentation schemes, especially broad ones such as AugMix, still show meaningful improvement on ImageNet-$\overline{\text{C}}$ over baseline. What we want to emphasize is that care must be taken when comparing methods, since the degradation differs dramatically for different augmentation schemes. Moreover, it does not appear exact augmentation-corruption overlap is especially predictive of poor generalization: AutoAugment includes exact overlap with some ImageNet-C corruptions but loses the least error compared to ImageNet-$\overline{\text{C}}$. This suggests it may not be sufficient to simply remove such augmentations from an augmentation scheme when testing on a corruption benchmark. Instead, it is important to understand more in-depth the interaction between an augmentation scheme and the tested corruptions. We hope that MSD and ImageNet-$\overline{\text{C}}$ may be an additional tools to help researchers understand why a robust augmentation scheme works and whether it may be expected to generalize to unknown corruptions.

## 6 DISCUSSION

***Corruption robustness as a secondary learning task.*** We have provided evidence that data augmentation can overfit a corruption benchmark. To explore this further, consider an analogy to a regular learning problem. We may think of corruption robustness in the presence of data augmentation as a sort of secondary task layered on the primary classification task: the set of data augmentations is the training set, the set of corruptions is the test set, and the goal is to achieve invariance of the underlying

Table 2: Test error for several data augmentation methods on CIFAR-10-$\overline{\text{C}}$ and ImageNet-10-$\overline{\text{C}}$. Every method performs worse on these new datasets than on ImageNet-C or CIFAR-10-C. Example corruptions and descriptions of the abbreviations are given in Appendix E, and standard deviations for individual corruption are given in Table 5 in Appendix D.2. 'Baseline' refers to default augmentation only. *ANT uses the single pre-trained model provided with the paper and has slightly different training parameters.

| Aug | IN-C Err | IN-$\overline{\text{C}}$ Err | $\Delta$IN-C | ImageNet-$\overline{\text{C}}$ Corruptions | | | | | | | | | |
|---|---|---|---|---|---|---|---|---|---|---|---|---|---|
| | | | | BSmpl | Plsm | Ckbd | CSin | SFrq | Brwn | Prln | ISprk | Sprk | Rfrac |
| Baseline | 58.2$_{\pm0.5}$ | 57.7$_{\pm0.2}$ | -0.5 | 68.6 | 71.7 | 49.4 | 84.7 | 79.0 | 37.5 | 34.3 | 32.4 | 76.7 | 42.8 |
| AA | 54.8$_{\pm0.2}$ | 55.7$_{\pm0.3}$ | +0.9 | 54.8 | 68.3 | 43.8 | 86.5 | 78.8 | 34.5 | 33.8 | 36.1 | 77.1 | 43.8 |
| PG | 48.1$_{\pm0.2}$ | 56.6$_{\pm0.4}$ | +8.5 | 60.3 | 74.1 | 48.5 | 82.1 | 76.7 | 38.9 | 34.6 | 32.1 | 76.5 | 42.1 |
| ANT* | 48.8 | 53.9 | +5.1 | 35.8 | 75.5 | 56.9 | 76.4 | 63.7 | 41.0 | 35.2 | 35.0 | 76.1 | 43.3 |
| AugMix | 49.1$_{\pm0.7}$ | 52.4$_{\pm0.2}$ | +3.3 | 43.2 | 72.2 | 46.1 | 76.3 | 67.4 | 38.8 | 32.4 | 32.3 | 76.4 | 39.2 |
| Aug | C10-C Err | C10-$\overline{\text{C}}$ Err | $\Delta$C10-C | CIFAR-10-$\overline{\text{C}}$ Corruptions | | | | | | | | | |
| | | | | BSmpl | Brwn | Ckbd | CBlur | ISprk | Line | P&T | Rppl | Sprk | TCA |
| Baseline | 27.0$_{\pm0.6}$ | 27.1$_{\pm0.5}$ | +0.1 | 42.9 | 27.2 | 23.3 | 11.8 | 43.3 | 26.2 | 11.3 | 21.6 | 21.0 | 42.9 |
| AA | 19.4$_{\pm0.2}$ | 21.0$_{\pm0.4}$ | +1.6 | 17.7 | 17.5 | 17.6 | 9.5 | 40.4 | 23.6 | 10.7 | 23.5 | 17.5 | 31.8 |
| PG | 17.0$_{\pm0.3}$ | 23.8$_{\pm0.5}$ | +6.8 | 9.0 | 30.1 | 21.6 | 12.8 | 35.4 | 20.6 | 8.8 | 21.5 | 19.3 | 59.5 |
| AugMix | 11.1$_{\pm0.2}$ | 16.0$_{\pm0.3}$ | +5.9 | 9.8 | 27.8 | 13.4 | 5.9 | 30.3 | 18.0 | 8.3 | 12.1 | 15.5 | 19.2 |

primary task. In this language, the 'datasets' involved are quite small: ImageNet-C has only 15 corruption types, and several augmentation schemes composite only around 10 basic transforms. To mitigate overfitting, standard machine learning practice would dictate a training/validation/test set split; it is only the size and breadth of modern vision datasets that has allowed this to be neglected in certain cases recently. But the effective dataset size of a corruption robustness problem is tiny, so having a held-out test set that is not used during model development seems necessary. To emphasize, this is not a test set of the underlying classification task, for which generalization has been studied in Recht et al. (2018; 2019). Instead, it is a test set of corruption transforms themselves. This means there would be two sets of dissimilar transformations, both applied to the ImageNet validation set, that would act as a validation/test split on transforms[2].

***Real-world corruption robustness.*** Recently, Hendrycks et al. (2020) and Taori et al. (2020) study how performance on corruption transforms generalizes to real-world corruptions and come to conflicting conclusions. Though we do not study real-world corruptions directly, we have proposed a mechanism that may explain the conflict: performance will generalize between transforms and real-world corruptions if they are perceptually similar, but will likely not if they are dissimilar. Since Hendrycks et al. (2020) and Taori et al. (2020) draw on different real-world and synthetic corruptions, it may simply be that the perceptual similarity between datasets differs in the two analyses. This also suggests a way to find additional corruption transforms that correlate with real-world corruptions: transforms should be sought that have maximal perceptual similarity with real-world corruptions.

***Generalization does occur.*** Finally, let us end on a positive note. Through our study of overfitting, we have encountered two features of data augmentation that may explain why it can be such a powerful tool for corruption robustness. First, within a class of perceptually similar transforms, generalization does occur. This means a single, simple data augmentation may confer robustness to many, much more complicated corruptions, as long as they share perceptual similarity. Second, the presence of dissimilar augmentations in an augmenation scheme often causes little to no loss in performance, as long as a similar augmentation is also present. We study this in a bit more detail in Appendix B by demonstrating that adding many dissimilar augmentations increases error much less than adding a few similar augmentations decreases it. Together, these features suggest broad augmentation schemes with many dissimilar augmentations may be capable of conferring robustness to a large class of unknown corruptions. More generally, we think data augmentation is a promising direction of study for corruption robustness, as long as significant care is taken to avoid overfitting.

---

[2]The validation set provided in Hendrycks & Dietterich (2018) consists of perceptually similar transforms to ImageNet-C and would not be expected to work well for the validation discussed here.

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

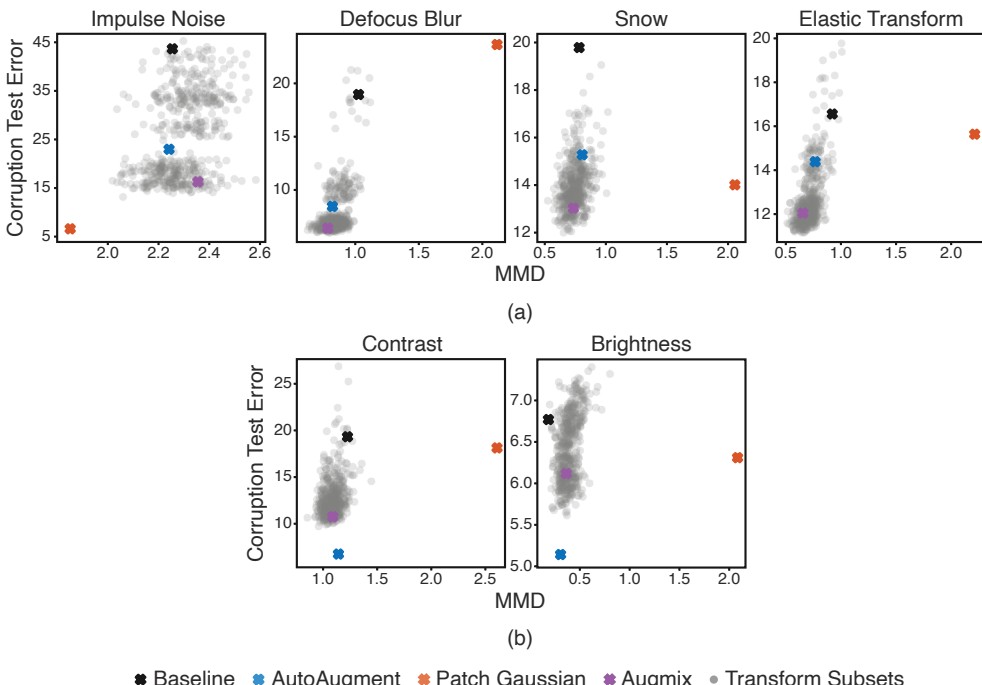

Figure 7: (a) Patch Gaussian shows a low MMD distance on the noise corruptions and a high MMD distance on every other corruption, suggesting that it overfits the noise corruptions. (b) While AutoAugment contains *contrast* and *brightness* augmentations, it is broad enough that it doesn't have a low MMD to these corruptions. Note that since *brightness* shows poor correlation for MSD, it is possible that in this case the MMD does not change for other reasons.

## A    ANALYZING OVERFITTING WITH MMD

Overfitting can be analyzed in a bit more detail by using the MMD distance from Section 3. Recall that low MMD may be indicative of overfitting a single type of corruption, since it suggests a possible equivalence between the augmentation and corruption distributions. Broad augmentation schemes will likely have low MMD with no single corruption distribution, but narrow ones that overfit to a single corruption will have low MMD with it and high MMD for others. Figure 7 shows example MMD-error correlations. For Patch Guassian, MMD is low for the noise corruptions and high for everything else, while AutoAugment and AugMix, which are constructed out of many visually distinct transforms, show no strong correlation. We might then expect greater overfitting from Patch Gaussian.

## B    SAMPLING SIMILAR AUGMENTATIONS MORE FREQUENTLY GIVES MINOR PERFORMANCE IMPROVEMENTS

Here we describe an alternative experiment that shows how the introduction of dissimilar augmentations affects corruption error. For a broad data augmentation scheme that provides robustness to many dissimilar corruptions, each corruption may only have a similar augmentation sampled some small fraction of the time. This small fraction of samples must be sufficient to yield good performance on each corruption to obtain robustness overall. We expect that this should be the case, since neural networks are often good at memorizing rare examples. Additionally, the toy problem in Figure 2 suggests that a large fraction of sampled augmentations may be dissimilar without significant loss in corruption error. Here we show the effect using a real augmentation scheme.

We consider performance on CIFAR-10-C when training with AugMix augmentations (we do not use their Jensen-Shannon divergence loss, which gives additional improvements). However, instead of sampling directly from the AugMix distribution during training, we first sample 100k transforms and

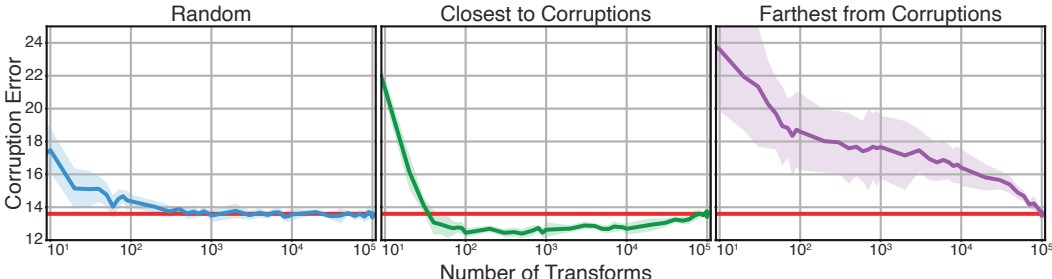

Figure 8: Average corruption error on ImageNet-C as a function the size of a fixed subset of AugMix augmentations. During training, augmentations are only sampled from the subset. The subset is chosen one of three ways: randomly, the most similar augmentations to ImageNet-C, or the least similar augmentations to ImageNet-C. Choosing similar corruptions improves error beyond AugMix, but not by as much that choosing dissimilar augmentations harms it.

sort these transforms by their distance to the CIFAR-10-C corruptions. This sorting is done to evenly distribute the augmentations among the 75 (15 corruptions in 5 severities) individual corruptions; *e.g.* the first 75 augmentations in the list are the closest augmentation to each corruption. Then we take a fixed-size subset $\mathbb{A}$ of these transforms and train on augmentations sampled only from this subset using the training parameters from Hendrycks et al. (2019). We select $\mathbb{A}$ three different ways: randomly, taking the $|\mathbb{A}|$ closest augmentations, and taking the $|\mathbb{A}|$ farthest augmentations. We then measure the average corruption error on CIFAR-10-C and plot this error against $|\mathbb{A}|$ in Figure 8.

First, we note that for randomly sampled augmentations, $\mathbb{A}$ does not need to be very large to match AugMix in performance. Even though training on AugMix with our training parameters would normally would produce 5 million uniquely sampled augmentations, only around 1000 are needed to achieve equivalent performance. Training on the closest augmentations exceeds regular AugMix performance with only around 100 unique transforms, which acts as additional evidence that augmentation-corruption similarity correlates with corruption error. This gain in accuracy comes not from having access to better transformations, but from having more frequent access to them at training time. However, the gain is fairly mild at only around 1%, even though the best transformations are sampled all of the time instead of rarely. The gain from frequency is much less than the gain from having more similar augmentations, since choosing the most dissimilar augmentations gives around a 5% drop in accuracy. This suggests that it is a net positive to decrease the frequency of sampling similar augmentations in order to include augmentations similar to another set of corruptions: the gain in accuracy on the new corruption set will likely out weight the small loss in accuracy on the original set.

## C  MSD Ablation

### C.1  Architecture choice

Here we provide evidence that changing the architecture of the feature extractor used in the definition of MSD does not have any qualitative effect on the correlation with corruption error. We use a version of VGG-19 with batch normalization that has been modified for CIFAR-10. Otherwise, all other parameters are chosen the same. We then repeat the experiment of Section 4. In Table 3 and Figure 9, we show that the qualitative results of this experiment are unchanged when using VGG-19-BN as the feature extractor.

### C.2  Parameter dependencies

In calculating the feature space for transforms and MSD, it is necessary to both pick a number of images to average over and a number of corruptions to average over. In our experiments, we use 100 images and 100 corruptions. Here we provide evidence that these are reasonable choices for these parameters.

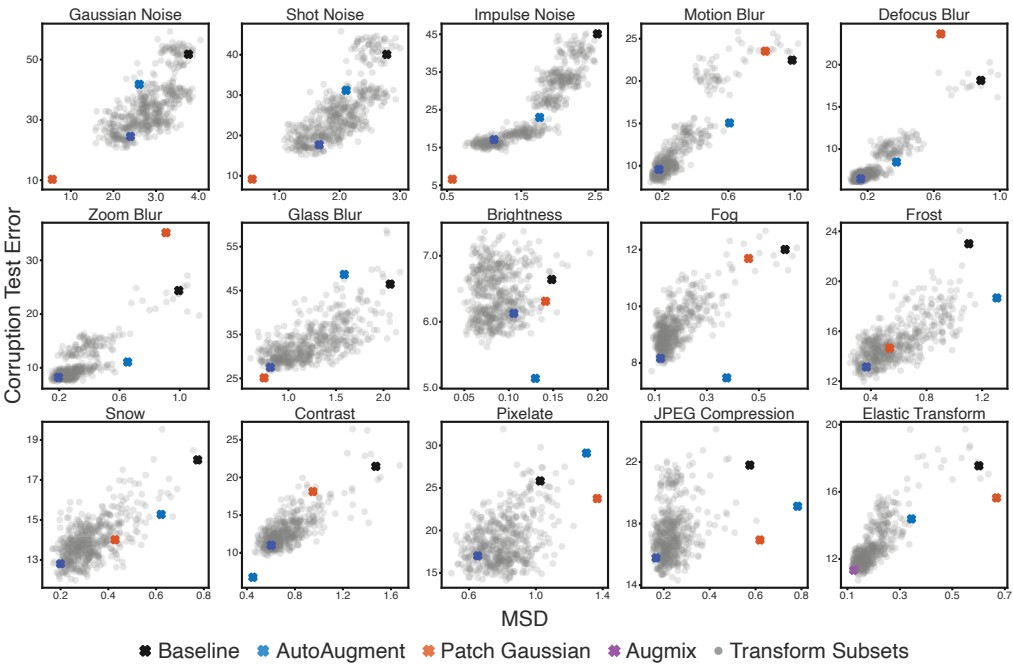

Figure 9: MSD vs corruption test error for which MSD is calculated using VGG-19-BN as the architecture for feature extraction. The corruption error is still calculated using WideResNet-40-2. Compare to Figure 16 to see that the qualitative structure of the correlation is the regardless of which architecture is used for the feature extractor.

Table 3: Spearman's rank coefficient for the correlation between MSD and corruption error for two architectures in the feature extractor: WideResNet-40-2 and VGG-19-BN. While WideResNet has slightly better correlations overall, the relative behavior across corruptions remains the same for the two architectures.

| Corruption | WRS | VGG | Corruption | WRS | VGG |
|---|---|---|---|---|---|
| Gaussian Noise | 0.76 | 0.70 | Fog | 0.68 | 0.60 |
| Shot Noise | 0.83 | 0.78 | Frost | 0.66 | 0.66 |
| Impulse Noise | 0.90 | 0.92 | Snow | 0.65 | 0.53 |
| Motion Blur | 0.86 | 0.81 | Contrast | 0.66 | 0.65 |
| Defocus Blur | 0.83 | 0.78 | Pixelate | 0.35 | 0.29 |
| Zoom Noise | 0.77 | 0.68 | JPEG Compression | 0.33 | 0.26 |
| Glass Blur | 0.69 | 0.66 | Elastic Transform | 0.77 | 0.74 |
| Brightness | 0.27 | 0.08 | | | |

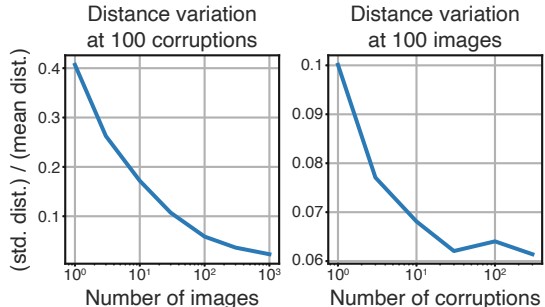

Figure 10: The standard deviation of the distance between an augmentation and a corruption center, taken over 100 resamplings of images and corruptions. The standard deviation is calculated as a percentage of the mean distance, then averaged over 100 augmentation-corruption pairs. At our choice of parameters, 100 images and 100 corruptions, the standard deviation is only around 5% of the distance. This is smaller than the feature size in the scatter plots of Figure 14

To do so, we use the augmentation scheme from AugMix and corruptions distributions from CIFAR-10-C to randomly sample 100 augmentation-corruption pairs. Then, for different samplings of a fixed number of images and sampled corruptions, we measure the augmentation-corruption distance in the transform feature space 100 times for each augmentation-corruption pair. We calculate the standard deviation of the distance as a percentage of the mean distance for each augmentation-corruption pair, and average this over pairs. The results are shown in Figure 10. For our choice of image and corruption number, the standard deviation in distance is only around 5% of the mean distance, which is smaller than the size of the features in the scatter plots in Figure 14.

## D IMAGENET-$\overline{\text{C}}$ DETAILS

### D.1 DATASET CONSTRUCTION DETAILS

First, 30 new corruptions, examples of which are shown in Figure 11, are adapted from common image filters and noise distributions available online (Huxtable, 2006; Gladman, 2016). These corruptions are generated in 10 severities such that the image remains human interpretable at all severities and the distribution of errors on a baseline model roughly matches that of ImageNet-C.

For each corruption, groups of 5 severities are generated that roughly match the average spread in error across severities in ImageNet-C on a baseline model. Seven of these groups are formed for each corruption, each with one of severity 3 through 8 as the center severity of the group of 5.

A candidate dataset is a set of 10 groups of severities, each from a different corruption whose average corruption error on a baseline model is within 1% of ImageNet-C. This is necessary so that a relative decrease in error of data augmented models is normalized against a fixed baseline. Also, more distorted, harder transforms are likely farther away, so if this wasn't fixed maximizing distance would likely just pick the hardest transforms in the highest severities. It was computationally infeasible to enumerate all candidate datasets, so they were sampled as follows. For each choice of 5 corruptions, one choice of severities was selected at random so that the average corruption error was within 1% of ImageNet-C, if it existed. Then random disjoint pairs of two sets of 5 were sampled to generate candidate datasets. 100k candidate datasets are sampled.

Call the set of all corruption-severity pairs in a dataset $\mathbb{C}$. The distance of a candidate dataset to ImageNet-C is defined as

$$d(\mathbb{C}_{\text{new}}, \mathbb{C}_{\text{IN}-\text{C}}) = \mathbb{E}_{c \sim \mathbb{C}_{\text{new}}} \left[ \min_{c' \sim \mathbb{C}_{\text{IN}-\text{C}}} d_{\text{MMD}}(c, c') \right], \tag{2}$$

where $d_{\text{MMD}}$ is defined in Section 3. The minimum helps assure that new corruptions are far from all ImageNet-C corruptions.

This distance is calculated for all 100k sampled candidate datasets. For CIFAR-10, the same parameters described in Section 4.1 are used to calculate the distance. For ImageNet, the feature

Table 4: Comparison between performance on ImageNet/CIFAR10-C and ImageNet/CIFAR10-C̄. Standard deviations are over 10 runs for CIFAR-10 and 5 runs for ImageNet. *ANT results use the pre-trained model provided with the paper and thus has slightly different training parameters and only one run.

| Aug | IN-C Err | IN-C̄ Err | ΔIN-C |
|---|---|---|---|
| Baseline | $58.2_{\pm0.5}$ | $57.7_{\pm0.2}$ | -0.5 |
| AA | $54.8_{\pm0.2}$ | $55.7_{\pm0.3}$ | +0.9 |
| PG | $48.1_{\pm0.2}$ | $56.6_{\pm0.4}$ | +8.5 |
| ANT* | 48.8 | 53.9 | +5.1 |
| AugMix | $49.1_{\pm0.7}$ | $52.4_{\pm0.2}$ | +3.4 |

| Aug | C10-C Err | C10-C̄ Err | ΔC10-C |
|---|---|---|---|
| Baseline | $27.0_{\pm0.6}$ | $27.1_{\pm0.5}$ | +0.1 |
| AA | $19.4_{\pm0.2}$ | $21.0_{\pm0.4}$ | +1.6 |
| PG | $17.0_{\pm0.4}$ | $23.8_{\pm0.5}$ | +6.8 |
| AugMix | $11.1_{\pm0.2}$ | $16.0_{\pm0.3}$ | +4.9 |

Table 5: Breakdown of performance on individual corruptions in ImageNet/CIFAR10-C̄. Standard deviations are over 10 runs for CIFAR-10 and 5 runs for ImageNet. Examples and full names of each corruption are given in Appendix E. *ANT results use the pre-trained model provided with the paper and thus has slightly different training parameters and only one run.

| Aug | ImageNet-C̄ Corruptions | | | | | | | | | |
|---|---|---|---|---|---|---|---|---|---|---|
| | BSmpl | Plsm | Ckbd | CSin | SFrq | Brwn | Prln | ISprk | Sprk | Rfrac |
| Baseline | $68.6_{\pm0.5}$ | $71.7_{\pm0.7}$ | $49.4_{\pm0.6}$ | $84.7_{\pm0.5}$ | $79.0_{\pm0.8}$ | $37.5_{\pm0.5}$ | $34.3_{\pm0.1}$ | $32.4_{\pm0.5}$ | $76.7_{\pm0.2}$ | $42.8_{\pm0.2}$ |
| AA | $54.8_{\pm0.7}$ | $68.3_{\pm0.7}$ | $43.8_{\pm1.0}$ | $86.5_{\pm0.6}$ | $78.8_{\pm0.9}$ | $34.5_{\pm0.8}$ | $33.8_{\pm0.2}$ | $36.1_{\pm1.0}$ | $77.1_{\pm1.2}$ | $43.8_{\pm0.2}$ |
| PG | $60.3_{\pm2.9}$ | $74.1_{\pm0.7}$ | $48.5_{\pm1.0}$ | $82.1_{\pm0.4}$ | $76.7_{\pm0.8}$ | $38.9_{\pm0.4}$ | $34.6_{\pm0.1}$ | $32.1_{\pm0.7}$ | $76.5_{\pm0.6}$ | $42.1_{\pm0.4}$ |
| ANT* | 35.8 | 75.5 | 56.9 | 76.4 | 63.7 | 41.0 | 35.2 | 35.0 | 76.1 | 43.3 |
| AugMix | $43.2_{\pm0.8}$ | $72.2_{\pm0.4}$ | $46.1_{\pm0.2}$ | $76.3_{\pm0.3}$ | $67.4_{\pm0.7}$ | $38.8_{\pm0.5}$ | $32.4_{\pm0.1}$ | $32.3_{\pm0.2}$ | $76.4_{\pm0.4}$ | $39.2_{\pm0.2}$ |

| Aug | CIFAR-10-C̄ Corruptions | | | | | | | | | |
|---|---|---|---|---|---|---|---|---|---|---|
| | BSmpl | Brwn | Ckbd | CBlur | ISprk | Line | P&T | Rppl | Sprk | TCA |
| Baseline | $42.9_{\pm5.1}$ | $27.2_{\pm0.5}$ | $23.3_{\pm0.6}$ | $11.8_{\pm0.4}$ | $43.3_{\pm0.8}$ | $26.2_{\pm0.9}$ | $11.3_{\pm0.3}$ | $21.6_{\pm1.2}$ | $21.0_{\pm1.1}$ | $42.9_{\pm2.7}$ |
| AA | $17.7_{\pm1.7}$ | $17.5_{\pm0.5}$ | $17.6_{\pm0.5}$ | $9.5_{\pm0.3}$ | $40.4_{\pm1.5}$ | $23.6_{\pm0.7}$ | $10.7_{\pm0.3}$ | $23.5_{\pm0.5}$ | $17.5_{\pm0.7}$ | $31.8_{\pm1.8}$ |
| PG | $9.0_{\pm1.1}$ | $30.1_{\pm1.1}$ | $21.6_{\pm0.8}$ | $12.8_{\pm0.5}$ | $35.4_{\pm1.6}$ | $20.6_{\pm0.5}$ | $8.8_{\pm0.2}$ | $21.5_{\pm0.9}$ | $19.3_{\pm0.5}$ | $59.5_{\pm3.5}$ |
| AugMix | $9.8_{\pm0.7}$ | $27.8_{\pm1.3}$ | $13.4_{\pm0.4}$ | $5.9_{\pm0.2}$ | $30.3_{\pm0.7}$ | $18.0_{\pm0.6}$ | $8.3_{\pm0.2}$ | $12.1_{\pm0.4}$ | $15.5_{\pm0.5}$ | $19.2_{\pm1.0}$ |

extractor is a ResNet-50 trained according to Goyal et al. (2017), except color jittering is not used as a data augmentation. Since there is much greater image diversity in ImageNet, we jointly sample 10k images and corruptions instead of independently sampling 100 images and 100 corruptions. Code for measuring distances and training models is based on pyCls (Radosavovic et al., 2019; 2020).

The corruptions are then ranked according the their average contribution to the dataset distance. This entire procedure is repeated 10 times for CIFAR and 5 times for ImageNet, and corruption contributions are averaged. The top 10 are chosen to form the new dataset. Of candidate datasets made up of these 10 corruptions, the one with baseline error closest to ImageNet-C is chosen, though run-to-run fluctuation still causes some variation.

### D.2 COMPLETE RESULTS

Here we show average results comparing ImageNet/CIFAR-10-C to ImageNet/CIFAR-10-C̄ in Table 4, and a breakdown of ImageNet/CIFAR-10-C̄ results by corruption in Table 5.

## E GLOSSARY OF TRANSFORMS

This appendix contains examples of the augmentations and corruptions discussed in the text. Figure 11 shows the 30 new corruptions introduced in Section 5. These transforms are adapted from common online filters and noise sources (Huxtable, 2006; Gladman, 2016). They are designed to be human interpretable and cover a wide range transforms, including noise additions, obscuring, warping, and color shifts. The 10 transforms chosen for ImageNet-C̄ are blue noise sample (BSmpl), plasma noise (Plsm), checkerboard (Ckbd), cocentric sine waves (CSin), single frequency (SFrq), brown noise (Brwn), perlin noise (Prln), inverse sparkle (ISprk), sparkles (Sprk), and caustic refraction (Rfrac). For CIFAR-10-C̄, there is blue noise sample (BSmpl), brown noise (Brwn), checkerboard (Ckbd), circular motion blur (CBlur), inverse sparkle (ISprk), lines (Line), pinch and twirl (P&T), ripple (Rppl), sparkles (Sprk), and transverse chromatic abberation (TCA).

Figure 12 shows the 9 base transforms used to build augmentation schemes in the analysis. These are transforms from the Pillow Image Library that are often used as data augmentation. They have no exact overlap with either the corruptions of ImageNet-C or the new corruptions we introduce here. There are five geometric transforms (shear x/y, translate x/y, and rotate) and four color transforms (solarize, equalize, autocontrast, and posterize). We choose this particular set of augmentations following Hendrycks et al. (2019).

Figure 13 shows example corruptions from the ImageNet-C benchmark (Hendrycks & Dietterich, 2018). They a grouped into four categories: noise (gaussian noise, shot noise, and impulse noise), blurs (motion blur, defocus blur, zoom blur, and glass blur), synthetic weather effects (brightness, fog, frost, and snow), and digital transforms (contrast, pixelate, JPEG compression, and elastic transform).

## F   SUPPLEMENTARY PLOTS

This appendix contains additional plots for the analysis in the main text.

Fig 14 shows a comparison of how MMD and MSD correlate with corruption error. MMD typically shows poor correlation, while MSD has strong correlation in all four categories of corruption.

Figure 15 shows the correlation between MSD and corruption error for all 15 ImageNet-C corruptions, where $\rho$ is the Spearman rank correlation. Here, 'AugMix' refers to just their augmentation scheme, and not their Jensen-Shannon divergence loss, which gives additional improvements in corruption error. 12 of 15 corruptions have a Spearman rank correlation greater than 0.6. The remaining three that show poor correlations are 'brightness', 'JPEG compression', and 'pixelate'.

Figure 16 shows the average contribution of a new corruption to the dataset's distance from ImageNet-C. The top 10 large average contributions, colored in blue, are chosen as the corruptions to make up the dataset ImageNet-C̄.

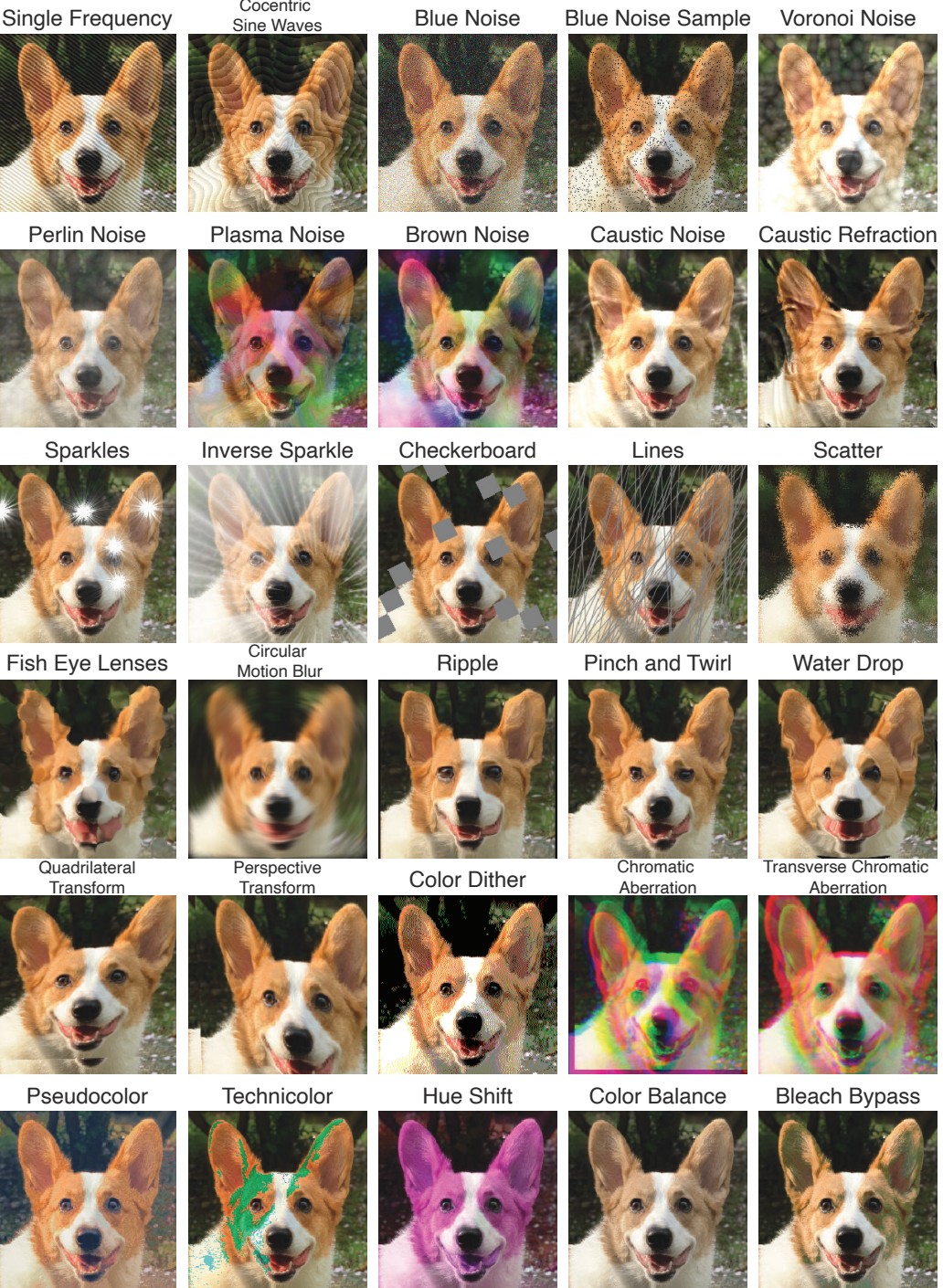

Figure 11: Examples of each corruption considered when building the dataset dissimilar to ImageNet-C. Base image © Sehee Park.

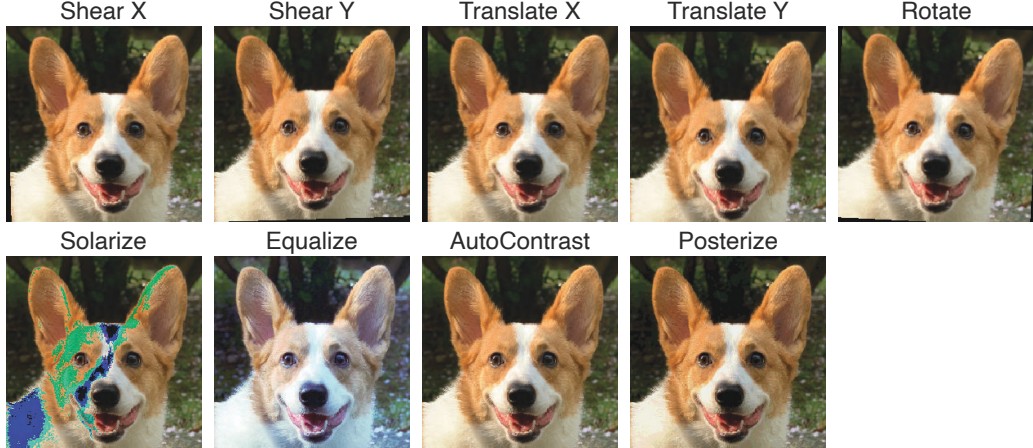

Figure 12: The nine base transforms used as augmentations in analysis. Base image © Sehee Park.

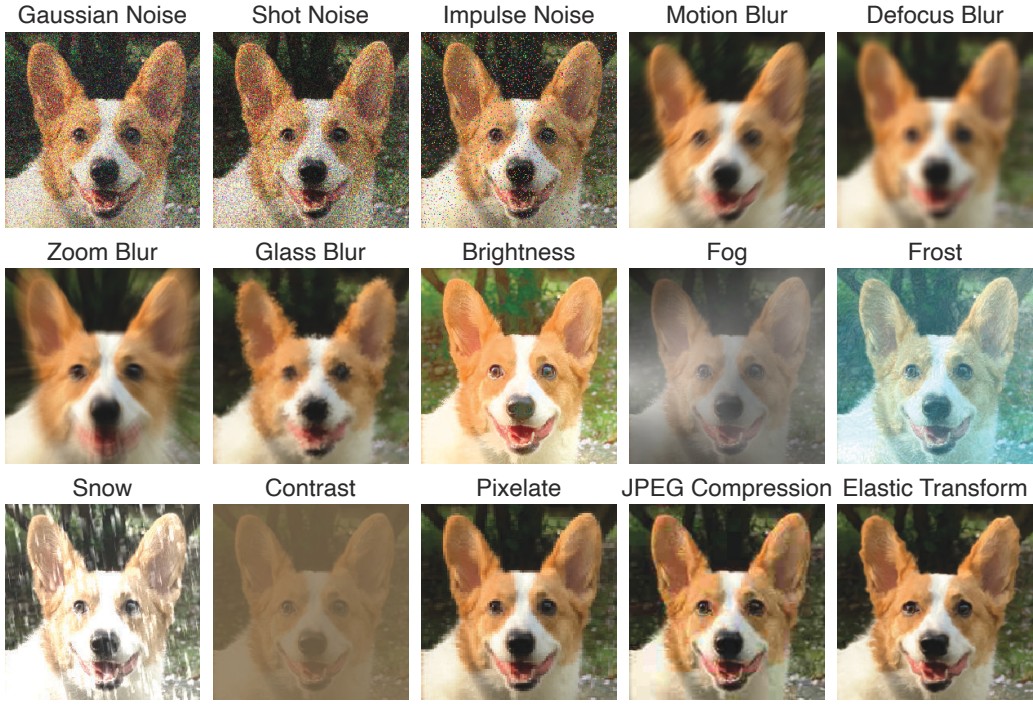

Figure 13: Examples of the 15 corruptions in the ImageNet-C corruption benchmark (Hendrycks & Dietterich, 2018). Base image © Sehee Park.

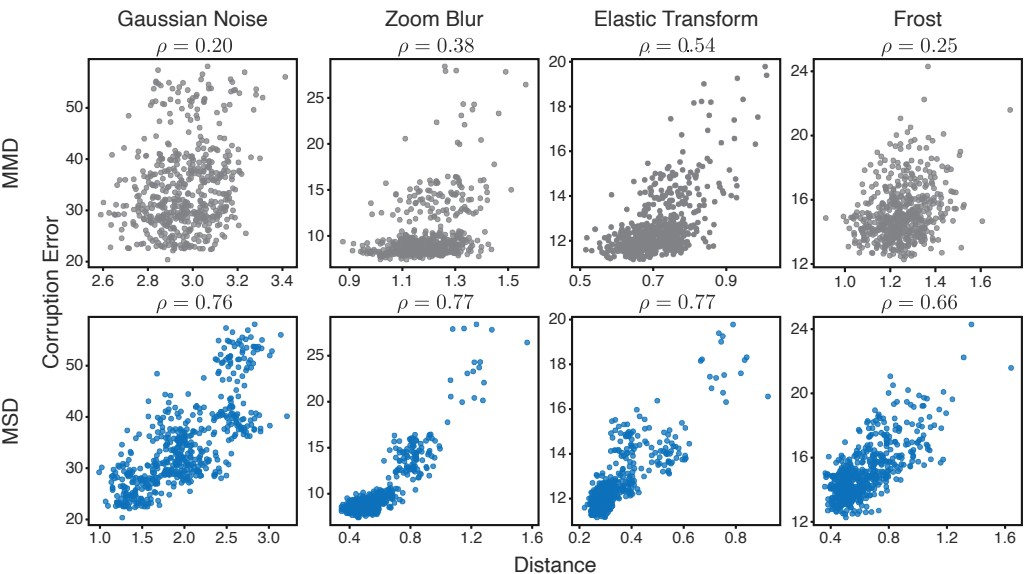

Figure 14: Example relationships between augmentation-corruption distance and corruption error for two distance scores, MMD and MSD. $\rho$ is the Spearman rank correlation. MMD between an augmentation and corruption distribution is not typically predictive of corruption error. MSD correlates well across all four categories of corruption in CIFAR-10-C.

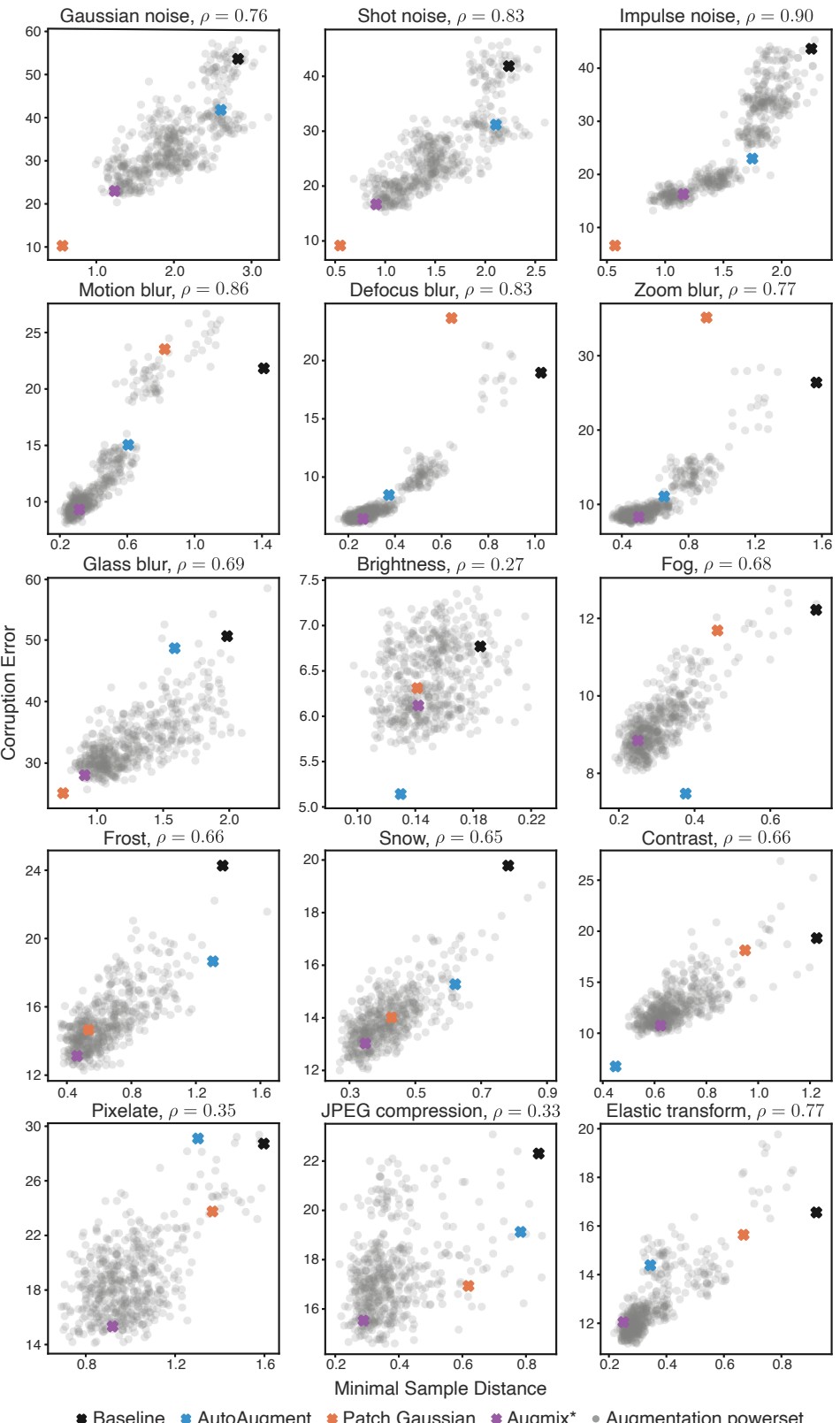

Figure 15: See text for details.

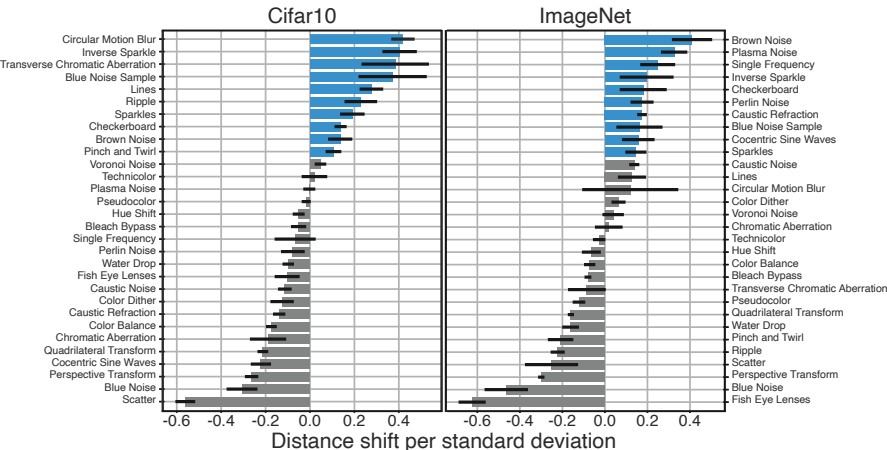

Figure 16: A corruption's average contribution to the distance to ImageNet-C, as a fraction of the population's standard deviation. The blue corruptions are those used to construct ImageNet-C̄.

