# OpenReview forum: "On interaction between augmentations and corruptions in natural corruption robustness"
_ICLR.cc/2021/Conference — Reject_

### Official Review · AnonReviewer3 · 2020-10-28
**"Is Robustness Robust?" It would appear so.**

**Rating:** 5
**Confidence:** 5

**Review:**

This paper proposes ImageNet-\bar{C} which uses a smaller number of carefully chosen corruptions, compared to ImageNet-C. The authors try to argue that previous work is overfitting to ImageNet-C. They claim "overfitting indeed occurs." Additionally, they propose "Minimum Sample Distance," showing that they can predict generalization performance using feature embedding distances.

I don't think they marshaled substantial, far-reaching evidence that "overfitting indeed occurs." They rendered numerous corruptions from Huxtable, 2006; Gladman, 2016 (which I very much appreciate) and adversarially chose the worst corruptions. As a result, the best models performed worse by a few percent compared to ImageNet-C.
_A drop is inevitable and expected given the adversarial selection. If the drop was consistently more than, say, 20%, then there might be strong evidence of overfitting._ Since the degradation for some techniques is small, this strikes me as evidence that by-and-large the community isn't overfitting. (Note it was evident to all that Patch Gaussian and ANT were specialized to noise corruptions.) "Is Robustness Robust?" It seems like the answer is "yes" but the authors are trying to argue that the answer is "no."

This paper is fairly similar to _Increasing the Coverage and Balance of Robustness Benchmarks by Using Non-Overlapping Corruptions_ (submission #1101). If these papers have very different ratings, then there's a problem with this review process.

"AugMix, which introduces a broad array of both geometric and color augmentations"
No it doesn't. It removes several color augmentations from AutoAugment and doesn't introduce any augmentation primitive beyond elementwise convex combinations.

"On other corruptions, the increase in error is even worse than the mean would suggest, and even broad augmentations like AugMix...."
AutoAugment is more broad; AugMix is narrower.

"Second, AutoAugment is the only tested augmentation scheme that was designed before the release of ImageNet-C"
AugMix uses a proper subset of AutoAugment's augmentations. It's not as though AugMix added in distortions to fit ImageNet-C; it removes augmentations because AA fits some of ImageNet-C's corruptions directly. This observation makes their overfitting case hard to maintain.

Also, if AutoAugment's full augmentation list is fair game for ImageNet-\bar{C}, then the authors should train AugMix with the full set of augmentations and make that comparison; the generalization discrepancy would likely be even smaller were the augmentation sets made equal.

From the "Corruption robustness as a secondary learning task" section:
"To mitigate overfitting, standard machine learning practice would dictate a training/validation/test set split; it is only the size and breadth of modern vision datasets that has allowed this to be neglected in certain cases recently."
"having a held-out test set that is not used during model development seems necessary."
"ImageNet-C has only 15 corruption types"
"ImageNet-C could serve as a validation set and ImageNet-\bar{C} could serve as a held-out test set"
Essentially, since the community lacks a validation set, ImageNet-C should become the validation set, and ImageNet-\bar{C} should become the test set.
This section might be negligent or worse for not acknowledging the already existent ImageNet-C validation set. ImageNet-C provides a validation set with about half the corruptions of ImageNet-\bar{C}. There are 19 available ImageNet-C corruptions, so the community already has a validation set.

The authors should cite or compare to other works that use feature distances to predict generalization. An example is "The Frechet Distance of training and test distribution predicts the generalization gap."

Update: "Noisy Student and Assemble-ResNet use without removing overlapping augmentations, yet they test on ImageNet-C." This is a bad practice and I should hope the authors of this paper only have experiments where there is a train-test mismatch (otherwise we're not testing robustness to distribution shift).

---

> ### Author Response · Authors · 2020-11-18
> **Response to reviewer 3**
>
> Thank you for your expert, in-domain reviews. It is not our intent to formulate our paper primarily as proposing a new corruption benchmark, nor to suggest that existing augmentation methods are not making meaningful progress on improving corruption robustness. ImageNet-C has enabled rapid progress that has already allowed for several new insights into how to improve corruption robustness, using both augmentations and other methods.
>
> However, approaches to augmentation-corruption overlap differ from paper to paper.  By developing a quantifiable measure for perceptual similarity, we aim to better understand how this overlap affects results. As additional methods are developed using benchmarks like ImageNet-C, we think it is important that scientists have concrete tools to determine why a method is working and if it should be expected to generalize beyond the tested benchmarks. For instance, recent work such as *The many faces of robustness: A critical analysis of out-of-distribution generalization* has shown that very broad, diverse augmentation distributions tend to generalize better to additional corruption benchmarks. Our work provides intuitions that support the case that such a broad augmentation method is needed.
>
> Thanks to your comments we realize that our title may prime a reader in a way that we did not anticipate and we will change it. Note, though, that we have already not mentioned ImageNet-C-Bar by name in the abstract to deemphasize the dataset as a primary result, and we will clarify the text to deemphasize it further. Our intent was that discussion of overfitting was a case study in how to use our tools and not a judgment on the field as a whole, and we will make this explicit.
>
> **A drop is inevitable and expected given the adversarial selection.** Note that no adversarial selection is performed with respect to the augmentations, so a drop in accuracy is not guaranteed. Only CIFAR-10/Imagenet-C was used in the process of finding dissimilar corruptions, and we deliberately did not look at augmentation results before fixing the dataset selection method. We will highlight this in the text. We want to point out that even the best methods don’t make a 20% improvement on ImageNet-C, so a 20% drop does not seem possible. We agree that the loss in error is small and that the methods studied here are still effective. However, we are arguing that care must be taken when comparing different augmentation methods. If a new method claims improvement over existing methods because of a small gain on ImageNet-C, then a small loss due to overfitting *is* important.
>
> **This paper is fairly similar to submission #1101.** We would draw a distinction with this submission, which does not study augmentations and appears to be arguing for flaws in ImageNet-C that require its replacement. We disagree with this point-of-view as we have already witnessed several new and exciting methods developed thanks to ImageNet-C. Instead, our goal is to provide tools to quantify how augmentation-corruption overlap affects corruption error, so that future methods developed using ImageNet-C can be better understood and compared.
>
> **AutoAugment is more broad; AugMix is narrower.** We would argue that, because AugMix can produce perceptually new effects by combining transforms (e.g. blur-like effects from superimposing differently shifted images, as shown in Figure 1), it may be broader than would be suggested by just counting individual base transforms. We will clarify this language.
>
> **The authors should train AugMix with the full set of augmentations.** We want to analyze these augmentation policies ‘as is’, since this is how they are often used in practice. This has already occurred with AutoAugment, which Noisy Student and Assemble-ResNet use without removing overlapping augmentations, yet they test on ImageNet-C.
>
> **Not acknowledging the already existent ImageNet-C validation set.** The validation set within ImageNet-C is quite different: these corruptions are perceptually similar to ImageNet-C corruptions, so we would not expect them to act as good validation for generalization to perceptually dissimilar corruptions. Since the goal is robustness to unknown image corruptions, generalization within perceptually similar transformations is unlikely to be sufficient. We will clarify these points and deemphasize ImageNet-C-Bar in this section.
>
> **The authors should cite or compare to other works that use feature distances to predict generalization.** Thank you for bringing this to our attention.  We note that for corruption robustness, it is not optimal to make the training distribution as close as possible to some given test distribution, since the ultimate goal is robustness to unknown natural corruptions. An important part of our approach is that we did not simply use an existing measure and that MSD addresses this difference. We will provide the recommended citations and clarify this difference in the revision.

---

### Official Review · AnonReviewer2 · 2020-10-28
**Review of AnonReviewer2**

**Rating:** 5
**Confidence:** 4

**Review:**

Summary
The paper studies the importance of similarity between augmentations and corruptions for improving performance on those corruptions. To measure the distance between the augmentation and corruption distributions, the paper proposes a new metric, Minimal Sample Distance (MSD), which is the perceptual similarity between an average corruption and the closest augmentation from a finite set of samples sampled from the augmented data distribution. It is shown that MSD overcomes the drawbacks of distributional distance measures like Maximum Mean Discrepancy (MMD). A new benchmark, called ImageNet-C-bar, made up of corruptions that are perceptually dissimilar to ImageNet-C, is introduced. Using standard evaluation, it is empirically shown that several recent augmentation schemes show degraded performance on the new dataset, suggesting that they generate augmentations only perceptually similar to ImageNet-C and thus are prone to overfitting.

+ves
+ Although the notion of the relation between augmentations and test-time corruptions has somewhat already been empirically observed and stated in many previous works, the paper tries to correlate this relation statistically. To my knowledge, this is the first such work.
+ Through comprehensive evaluations, the paper shows that the proposition of computing MSD rather than MMD correlates well with the relation between augmentations and corruptions observed in reality.
+ A new benchmark, ImageNet-C-bar is proposed, which shows a useful result to the community that recent SOTA augmentation methods have degraded performance on the new dataset because they generate augmentations close to ImageNet-C corruptions.

Concerns
- The paper says (pg 2) - “we empirically show an intuitive yet surprisingly overlooked finding: Augmentation-corruption perceptual similarity is a strong predictor of corruption error”. However, this notion of the relation between augmentations and test-time corruptions does not seem very surprising. It’s perhaps well-known that DNNs will generalize well only when test distributions are fairly similar to training distributions. Hence, the importance of this observation does seem to be overemphasized, although this is certainly of use.

- It's been recently shown that removing texture biases by introducing stylized transformations also improves robustness to common-corruptions (Geirhos et al, ICLR 2019). However, stylized transformations don’t look close to any Imagenet-C corruptions. If MSD of stylized transforms is large (which intuitively seems so), then it will mean that MSD is not a reliable metric in such a case. Was this studied? Would MSD be a reliable metric even in such cases? It would be good to understand this, to get a more well-rounded picture of the proposed metric.

- Paper introduces MSD as a distance metric. However, distance metrics should be symmetric in nature. It is not quite evident from just Eq-1 if MSD is symmetric.

- In addition, adding high severity corruption as training augmentation leads to better performance on the same low severity corruption but vice-versa is not true in general. This suggests that measures should perhaps ideally be asymmetric. Clarifying the notion of “distance metric” in this work may be important to make the work mathematically correct.

- Standard choices for measuring perceptual distances are VGG-16 or 19 networks pre-trained on ImageNet. However, the paper chooses to use WRN-40-10 trained on CIFAR-10. This seems to deviate from standard settings. The paper should explain the rationale behind their choice. It will be great to show an ablation on how this choice of feature extractor (VGG-19, Robust VGG etc) affects the MSD. Ideally, the metric should be robust i.e. shouldn’t be sensitive to the choice of feature extractor.

- The practical utility of ImageNet-C-bar seems limited and unclear. The only use that I can think of is using it to identify overfitting on ImageNet-C. I would be happy to understand what I am missing here.

POST-REBUTTAL:

I thank the authors for the response and the revisions to the paper. I appreciate the authors' efforts towards the rebuttal. I am however left with some concerns which did not have a convincing resolution:

* Regarding the comment on how the proposed analysis would look for stylized transforms, the authors say in the response that "...we don’t expect that perceptual similarity is the only cause of improved corruption error, only that perceptual similarity is particularly salient for predicting generalization to dissimilar corruptions...". The work seems to be one-sided in this regard. If stylized transforms don't look perceptually similar to ImageNet-C corruptions but provide robustness, this counters the proposed hypothesis. It then becomes important to say where the proposed analysis is meaningful and where it is not. This seems to be lacking at this time in the work.

* Regarding the robustness of MSD to the choice of feature extractor (as also asked by R1), the revised paper includes results on VGG as feature extractor (thanks to the authors for this), but uses a model that is finetuned for CIFAR-10. In general, perceptual similarity is studied directly taking VGG pre-trained on ImageNet - without finetuning on the target dataset. This leaves this question open, and makes one wonder if the latter features did not support the analysis.

* The utility of Imagenet-C-bar as an additional benchmark to check the goodness of performance on Imagenet-C seems a bit convoluted. Would we need a Imagenet-C-bar-bar to check the goodness for corruptions that may be beyond perceptual similarity (such as stylized transforms)? This is not very convincing.

Overall, I am still on the fence on this work (and retain my original decision at this time). I think the paper does present an interesting insight, but I am not very convinced it has been studied and explored comprehensively enough. I would have ideally preferred to give a borderline (neither positive nor negative) decision, and will not be disappointed if the work is accepted, considering the interesting insights it offers.

---

> ### Author Response · Authors · 2020-11-18
> **Response to reviewer 2**
>
> Thank you for recognizing the novelness of quantifying the relationship between perceptual similarity and corruption error, as well as the value of MSD over MMD. You raise several excellent points that we address below.
>
> **This notion of the relation between augmentations and test-time corruptions does not seem very surprising.** We agree that the relation between augmentations and test-time corruptions is not unknown, but what we find surprising is that it is nevertheless often overlooked in analyses.  This is what we intended to emphasize in stating “we empirically show an intuitive yet **surprisingly overlooked** finding”. In particular, the approaches used to address the problem are usually ad hoc or missing entirely, and often differ substantially from paper to paper. For example, AugMix removes individual augmentations with exact overlap; Patch Gaussian is perceptually similar to the noise corruptions, but instead of modifying the augmentation tests on a subset of corruptions; others still, such as Noisy Student or Assemble-ResNet, use AutoAugment without removing overlapping augmentations but still report ImageNet-C results. This makes it somewhat challenging to fairly compare different robustness-improving methods. By quantifying the relationship and defining an explicit measure of augmentation-corruption similarity, our goal is to provide a tool to better answer ‘why does this work?’ and ‘will this generalize beyond the tested benchmarks?’ for methods proposed in the future.
>
> **If MSD of stylized transforms is large (which intuitively seems so), then it will mean that MSD is not a reliable metric in such a case. Was this studied?** We agree SIN would provide an interesting perspective on our method. Unfortunately, there is no CIFAR-10 version of SIN, and it is too computationally expensive to perform the analysis of section 4 on ImageNet. However, if SIN had large MSD despite its good performance, it could be interpreted as a good sign for SIN: if it performs well on a benchmark without being perceptually similar to it, one might expect that it is more likely to generalize to other perceptually dissimilar corruptions as well. More generally, we don’t expect that perceptual similarity is the only cause of improved corruption error (e.g., Yin et al 2019 study the importance of frequency dependence), only that perceptual similarity is particularly salient for predicting generalization to dissimilar corruptions.  We will reinforce this perspective in the paper.
>
> **This suggests that measures should perhaps ideally be asymmetric.** Your intuition regarding an asymmetric measure of perceptual similarity is correct, and MSD is indeed asymmetric. We mention this in section 3 when comparing to the symmetric MMD measure.  Thank you for pointing out the improper usage of the word ‘metric’ to describe MSD: it is a distance in the colloquial sense and not in the formal metric sense, and we did not mean to imply MSD is symmetric. We will remove reference to a metric in the revision.
>
> **The metric should be robust i.e. shouldn’t be sensitive to the choice of feature extractor.**  We have chosen to use ResNet since it is widely used for robustness benchmarks in the literature, so we reuse it for simplicity. We agree the measure should be robust to changes in the architecture of the feature extractor. We have results using VGG (modified for CIFAR-10) as the feature extractor but see no qualitative changes in how different corruptions and augmentations correlate. We will add plots for this in the revision and have included a table of correlations below for comparison.
>
> **The practical utility of ImageNet-C-bar seems limited and unclear.** At the moment, ImageNet-C is the most common benchmark for studying corruption robustness, and we expect that it will remain so since it is effective and simple to use. Due to the issues comparing methods discussed in the first paragraph above, we think having an independent check on ImageNet-C performance is itself valuable. More generally, our measure of perceptual similarity could be applied to generate dissimilar transforms for future benchmarks. In this sense, it is a general purpose tool for studying if a method will generalize beyond a tested corruption benchmark.
>
> **Comparison of Spearman’s coefficient for different feature extractors.**
>
> |Corruption | WRS-40-2 &nbsp; | VGG-19-BN &nbsp; |
> |---------|-------|------|
> | Gaussian Noise | 0.76 | 0.71 |
> | Shot Noise | 0.83 | 0.78 |
> | Impulse Noise | 0.90 | 0.92 |
> | Motion Blur | 0.86 | 0.81 |
> | Defocus Blur | 0.83 | 0.78 |
> | Zoom Blur | 0.77 |0.68 |
> | Glass Blur | 0.69 | 0.66 |
> | Brightness | 0.27 | 0.08 |
> | Fog | 0.68 | 0.60 |
> | Frost | 0.66 | 0.66 |
> | Snow | 0.65 | 0.53 |
> | Contrast | 0.66 | 0.65 |
> | Pixelate | 0.35 | 0.29 |
> | JPEG Compression &nbsp; | 0.33 | 0.26 |
> | Elastic Transform | 0.77 | 0.74 |

---

### Official Review · AnonReviewer4 · 2020-10-28
**Interesting analysis, but some strange comparisons**

**Rating:** 6
**Confidence:** 4

**Review:**

The paper introduces the Minimal Sample Distance (MSD): a measure of the minimal distance, in a trained network representation space, between samples modified with an augmentation and the average of all samples modified by a corruption. It uses this metric to claim that there exists a high correlation between the corruption error and the MSD of a given augmentation. This way, it claims that focusing on benchmarks like ImageNet-C may lead to overfitting to the corruptions present in that benchmark.

One problem is that this correlation isn’t true for all augmentations. Only 4 are highlighted in the main text. As the paper describes, a few corruptions have spearman coefficient of less than 0.5. Particularly notable is brightness which, despite having very low spearman coefficient, is used to show that Patch Gaussian has “overfit” to the noise corruptions in the ImageNet-C benchmark. This is especially worrying since in their original paper, Patch Gaussian shows improvement in non-noise as well, which couldn’t have come from overfitting. Additionally, why was AutoAugment, but not RandAugment tested?

Another concern is that it may not make sense to compare single augmentations (such as Patch Gaussian) with augmentation policies, such as AutoAugment, RandAugment, and AugMix. It’s possible that individual augmentations in these policies “overfit” to the corruptions as well, but that this isn’t shown in MSD due to the use of many corruptions. In which case, using PatchGaussian in the RandAugment search space (for instance) would resolve any issues.

The paper argues that AutoAugment "overfits" while AugMix doesn't, but that's only because AugMix explicitly removed any augmentations in ImageNet-C from its search space, so it would make sense to repeat this experiment with the augmentations present in AugMix in order to confirm that it doesn't indeed "overfit".

The paper then suggests that one solution would be to use MSD to sample dissimilar corruptions to test on. However, given that it seems like there’s no evidence of overfitting for augmentation policies that encourage a diversity of augmentations, such as AugMix (which is in line with Yin et al 2019). Then I’m not sure what the benefit of expanding the robustness benchmark is. If current methods have indeed overfit, why won’t we also overfit to the new benchmark’s corruptions?

Overall, the paper presents interesting results and discussion.

Update after rebuttal: I appreciate the authors' response and clarifications. I maintain my original score.

---

> ### Author Response · Authors · 2020-11-18
> **Response to reviewer 4**
>
> Thank you for recognizing the value of our work that quantifies augmentation-corruption similarity and measures generalization to perceptually dissimilar corruptions. You raise several important points that we address below.
>
> **This correlation isn’t true for all augmentations.** It is true that the correlation between MSD and error does not hold in all cases. We do not think this is surprising: perceptual similarity is only one of multiple interactions between augmentations and corruptions (e.g., the frequency dependence of Yin et al 2019 is another). One of the goals of this work is to add to the set of tools researchers can use to answer ‘why does this work?’ and ‘will this generalize beyond the tested benchmarks?’ for future robustness-improving methods. MSD is one such tool.  We agree this argument should be more explicit and will provide a table of all correlations in the main text so that it is clearer how many correlate well.
>
> **The paper argues that AutoAugment "overfits" while AugMix doesn't.** Thank you for pointing out that our language at the end of section 4 is confusing: we do not actually intend to imply this. Indeed, at the bottom of page 7, we argue that despite including exact overlaps with ImageNet-C, AutoAugment actually generalizes the best, since it shows the least degradation on the new corruptions. Section 4 seeks to establish that our measure captures intuitively meaningful information about perceptual similarity, including both exact overlaps like in AutoAugment and perceptually similar ones like for AugMix and Patch Gaussian. We will add additional clarification of this point. Furthermore, in this context our goal is not to argue that any single method like Patch Guassian is unuseful. We agree that Patch Gaussian might be successfully combined with a policy like RandAugment, and that Patch Gaussian may be performant on some corruptions for reasons other than perceptual similarity.
>
> **Brightness is used to show that Patch Gaussian has “overfit” to the noise corruptions in the ImageNet-C benchmark.** We do not make this argument in this way.  In Appendix A, we discuss MMD, for which the Spearman coefficient is low for all corruptions (this is not MSD, which we use in the main text). Our argument here needs only that Patch Gaussian’s MMD is much lower for these corruptions than for other types. This suggests Patch Gaussian is perceptually similar to only the noise corruptions.  Broader augmentation policies, which have components perceptually similar to many dissimilar corruptions, will have low MMD to none of them. Brightness and contrast are included as examples of this second statement, since AutoAugment contains these transforms explicitly but does not have lower MMD here than other augmentations. The behavior of MMD for Patch Gaussian on brightness explicitly is not critical for our argument: it is large for all non-noise corruptions. We will clarify this in the text of Appendix A and include additional plots.
>
> **Why was AutoAugment, but not RandAugment tested?** We benchmark AutoAugment because many robustness-improving augmentation papers use AutoAugment but not RandAugment as a component or baseline. We are working on RandAugment, but are having reproducibility problems (similar to github.com/ildoonet/pytorch-randaugment). We will continue working on it and try our best to include results in the revision. Since RandAugment is designed to improve search efficiency but otherwise uses the same individual transforms and compositing method as AutoAugment, we expect similar results.
>
> **It may not make sense to compare single augmentations (such as Patch Gaussian) with augmentation policies.** MSD is designed to handle both individual augmentations and policies over many augmentations, and thus it makes sense to apply it to both. We want a unified approach that can be used to study future augmentation strategies even if there is not a clear division into a policy over separate individual augmentations. Our results also provide quantifiable evidence supporting the intuition you suggest here: broad augmentation policies do appear to perform better than single augmentations.
>
> **It seems like there’s no evidence of overfitting for augmentation policies that encourage a diversity of augmentations.** While diverse augmentation policies perform much better, we do not agree that there is no evidence of overfitting. As noted in section 5, there are several new corruptions that AugMix and AutoAugment perform no better than baseline on.
>
> **If current methods have indeed overfit, why won’t we also overfit to the new benchmark’s corruptions?** We agree this could occur. However, it is scientifically advantageous if researchers are aware of the problem and have the tools to measure it. Additionally, our method of finding dissimilar corruptions could be applied again, were there need to identify poor generalization beyond our own or future datasets.

---

### Official Review · AnonReviewer1 · 2020-10-30

**Rating:** 7
**Confidence:** 4

**Review:**

The paper introduces a metric to quantify the perceptual similarity between different kind of corruptions and uses it to show that training on a corruptions induces robustness to other corruptions which are perceptually similar. Analyzing the current data augmentation based methods for robustness using this lens, the authors hypothesis that we are currently overfitting to the existing robustness benchmark (ImageNet-C) and proposes a new benchmark with a new set of corruptions.

I think the author's observations that training on some corruptions helps the network to be robust to similar corruptions in test time is quite intuitive. I appreciate that the paper tries to quantify this and performs an extensive empirical study. The insight from this study and the new proposed benchmark will be useful to further advance the research in this area.

I have a some questions/clarifications:
1. What are the augmentations that were used while training the network used to extract features for the metric? Does the metric will probably depend on the architecture of the network used as well. Have you verified that the metric is robust to different architectures and the same conclusion holds? I am not sure if this is in the supplementary material somewhere and I missed it.
2. Do we find any non-intuitive pairs of corruptions which are similar? It occurs to me that most geometric based corruptions are similar and noise based corruptions are similar etc, but is there any pair of corruptions across these groups that the metric identifies as similar?


Update after rebuttal: I appreciate the author response. I will maintain my original score.

---

> ### Author Response · Authors · 2020-11-18
> **Response to reviewer 1**
>
> Thank you for your excellent comments and questions.  We fully agree that the result is intuitive, yet published works often ignore the intuition and remove only augmentations that exactly match corruptions.  In particular, corruption robustness provides a new and exciting task, and we expect increased effort to improve performance on benchmarks such as ImageNet-C.  It will be very important that future reviewers and researchers have quantitative tools on hand to accurately judge and understand these proposals.  Regarding your questions:
>
> **What are the augmentations that were used while training the network used to extract features for the metric?** The augmentations used to train the feature extractor are the default ones for CIFAR-10 or ImageNet.  This is random crop and horizontal flip for CIFAR-10, and random crop, resize, and horizontal flip for ImageNet (e.g., as used by torchvision for training models https://github.com/pytorch/vision).
>
> **Have you verified that the metric is robust to different architectures and the same conclusion holds?** We fully agree that the ability to work with different feature extractors is crucial for the framework. We have results using VGG (modified for CIFAR-10) as the feature extractor and find no qualitative differences in how different corruptions and augmentations correlate.  We will add plots for this in the revision and have included a table of correlations below for comparison.
>
> **Do we find any non-intuitive pairs of corruptions which are similar?** Yes! One such example is presented at the bottom of page 6.  We found glass blur acts more like a noise corruption than a blur corruption.  This may be because the glass blur algorithm involves permuting pixel values spatially in addition to gaussian blurring.  Another is that the contrast corruption is improved by geometric augmentations more than by equalize or auto-contrast augmentations, which are explicitly contrast changing.  This may be because equalize and auto-contrast increase contrast, while the contrast corruption and blurring effects from superimposing geometric augmentations decrease contrast.  This is an example of the danger of relying on naming or intuition to remove an overlapping augmentation: naively auto-contrast and contrast would be very similar, but we actually find they behave quite differently!  We expect MSD will help researchers understand potential overlaps in a much more concrete and unified way.
>
>
> **Comparison of Spearman's coefficient for different feature extractors.**
>
> |Corruption | WRS-40-2 &nbsp; | VGG-19-BN &nbsp; |
> |---------|-------|------|
> | Gaussian Noise | 0.76 | 0.71 |
> | Shot Noise | 0.83 | 0.78 |
> | Impulse Noise | 0.90 | 0.92 |
> | Motion Blur | 0.86 | 0.81 |
> | Defocus Blur | 0.83 | 0.78 |
> | Zoom Blur | 0.77 |0.68 |
> | Glass Blur | 0.69 | 0.66 |
> | Brightness | 0.27 | 0.08 |
> | Fog | 0.68 | 0.60 |
> | Frost | 0.66 | 0.66 |
> | Snow | 0.65 | 0.53 |
> | Contrast | 0.66 | 0.65 |
> | Pixelate | 0.35 | 0.29 |
> | JPEG Compression &nbsp; | 0.33 | 0.26 |
> | Elastic Transform | 0.77 | 0.74 |

---

### Decision · Program_Chairs · 2021-01-07
**Final Decision**

**Decision:**

Reject

**Comment:**

The paper investigates the relationship between data augmentations used during training and their effect on the accuracy when evaluated on unseen corruptions at test-time. The paper proposes a metric called minimal sample distance (MSD) to measure the similarity between augmentations during training time and corruptions at test time.

The reviewers agree that the paper aims to solve an important problem and the paper has some interesting findings. However, the current version has a few shortcomings:
- Some of the claims about “overfitting” are confusing, especially for data augmentations that use ops similar to those in ImageNet-C. This is already known and which is why some papers uses a subset of operations (e.g. AugMix uses a subset of AutoAugment operations).
- The main take-home message and novelty is unclear: The initial version titled (“Is Robustness Robust?“) seemed to argue that we may be overfitting to Imagenet-C, but the rebuttal and the updated version revised some of the claims (see response to R3 and R4).  In light of the revision, I’m not sure how the main take-home messages differ from existing papers such as Yin et al. 2019 or “Many faces of robustness”.
- One of the main differences is quantification of the distribution similarity, however, as pointed out by R2, this analysis does not explain when stylized corruptions would help, so the current version of the paper feels a bit incomplete to me.

I recommend the authors to revise the draft based on reviewer feedback and resubmit the paper to another venue.